# Expanding the CRISPR Toolbox for Engineering Lycopene Biosynthesis in *Corynebacterium glutamicum*

**DOI:** 10.3390/microorganisms12040803

**Published:** 2024-04-16

**Authors:** Zhimin Zhan, Xiong Chen, Zhifang Ye, Ming Zhao, Cheng Li, Shipeng Gao, Anthony J. Sinskey, Lan Yao, Jun Dai, Yiming Jiang, Xueyun Zheng

**Affiliations:** 1Key Laboratory of Fermentation Engineering (Ministry of Education), National “111” Center for Cellular Regulation and Molecular Pharmaceutics, Cooperative Innovation Center of Industrial Fermentation (Ministry of Education & Hubei Province), School of Life and Health Sciences, Hubei University of Technology, Wuhan 430068, China; 102100541@hbut.edu.cn (Z.Z.); cx163_qx@163.com (X.C.); 102210707@hbut.edu.cn (Z.Y.); yaolislan1982@aliyun.com (L.Y.); jimdai913@sina.com (J.D.); 2110511321@hbut.edu.cn (Y.J.); 2Department of Pharmaceutical Chemistry, School of Pharmacy, The University of Kansas, Lawrence, KS 66047, USA; mingzhao@ku.edu; 3Department of Biology, Massachusetts Institute of Technology, Cambridge, MA 02139, USA; chengl@mit.edu (C.L.); asinskey@mit.edu (A.J.S.); 4School of Food and Biological Engineering, Jiangsu University, Zhenjiang 212013, China; shipeng.gao@ujs.edu.cn

**Keywords:** *Corynebacterium glutamicum*, CRISPR/MAD7, metabolic engineering, lycopene production

## Abstract

Lycopene represents one of the central compounds in the carotenoid pathway and it exhibits a potent antioxidant ability with wide potential applications in medicine, food, and cosmetics. The microbial production of lycopene has received increasing concern in recent years. *Corynebacterium glutamicum* (*C. glutamicum*) is considered to be a safe and beneficial industrial production platform, naturally endowed with the ability to produce lycopene. However, the scarcity of efficient genetic tools and the challenge of identifying crucial metabolic genes impede further research on *C. glutamicum* for achieving high-yield lycopene production. To address these challenges, a novel genetic editing toolkit, CRISPR/MAD7 system, was established and developed. By optimizing the promoter, ORI and PAM sequences, the CRISPR/MAD7 system facilitated highly efficient gene deletion and exhibited a broad spectrum of PAM sites. Notably, 25 kb of DNA from the genome was successfully deleted. In addition, the CRISPR/MAD7 system was effectively utilized in the metabolic engineering of *C. glutamicum*, allowing for the simultaneous knockout of *crtEb* and *crtR* genes in one step to enhance the accumulation of lycopene by blocking the branching pathway. Through screening crucial genes such as *crtE*, *crtB*, *crtI*, *idsA*, *idi*, and *cg0722*, an optimal carotenogenic gene combination was obtained. Particularly, *cg0722*, a membrane protein gene, was found to play a vital role in lycopene production. Therefore, the CBIEbR strain was obtained by overexpressing *cg0722*, *crtB*, and *crtI* while strategically blocking the by-products of the lycopene pathway. As a result, the final engineered strain produced lycopene at 405.02 mg/L (9.52 mg/g dry cell weight, DCW) in fed-batch fermentation, representing the highest reported lycopene yield in *C. glutamicum* to date. In this study, a powerful and precise genetic tool was used to engineer *C. glutamicum* for lycopene production. Through the modifications between the host cell and the carotenogenic pathway, the lycopene yield was stepwise improved by 102-fold as compared to the starting strain. This study highlights the usefulness of the CRISPR/MAD7 toolbox, demonstrating its practical applications in the metabolic engineering of industrially robust *C. glutamicum*.

## 1. Introduction

Lycopene is a lipid-soluble natural carotenoid pigment that exhibits the highest antioxidant ability compared to other carotenoids, which are supposed to prevent cancer and enhance health status. Thus, it finds extensive applications in medicine, food, and cosmetics [1,2]. Lycopene can be obtained through direct extraction from natural sources. While products extracted by this method possess high quality and biological activity, the extraction process proves relatively costly due to the low lycopene content. Chemical synthesis offers a cost advantage with low raw material expenses, but it raises concerns about health and food safety [3]. Consequently, constructing microbial cell factories for the production of lycopene emerges as an appealing alternative as it enables the complete synthesis of lycopene from inexpensive carbon sources. In addition, this approach holds the potential to enhance yield, promote sustainability, and reduce production costs. In the process of microbial biosynthesis, lycopene is composed of seven isopentenyl diphosphates (IPP) and one dimethylallyl diphosphate (DMAPP) [4], which are supplied by the mevalonate (MVA) pathway or the methylerythritol 4-phosphate (MEP) pathway. IPP is converted reversibly to DMAPP by isopentenyl diphosphate isomerase (IDI). Subsequently, IPP and DMAPP were condensed to generate farnesyl diphosphate (FPP) [5], which is then catalyzed via farnesyl diphosphate synthase (FPPS), geranylgeranyl diphosphate synthase (GGPPS, CrtE/IdsA), phytoene synthase (CrtB), and phytoene desaturase (CrtI) to form lycopene.

*Corynebacterium glutamicum* is an ideal and secure strain for the industrial production of numerous compounds [6], including amino acids, vitamins, terpenoids, and biofuels [7]. Compared with *E. coli* and yeast, *C. glutamicum* has various advantages, such as the utility of a variety of carbon sources, high rates of sugar consumption under both aerobic or anaerobic conditions, high tolerance to osmotic pressure and various chemicals, and the absence of endotoxin. Moreover, *C. glutamicum* possesses the *crtE*, *crtB*, and *crtI* genes that integrate the lycopene production pathway, utilizing precursors from the MEP pathway. Consequently, there is no requirement for introducing heterologous genes. This capability makes *C. glutamicum* a suitable chassis cell for lycopene production, underscoring its potential feasibility [8]. Nonetheless, certain genes have been identified to divert lycopene towards the production of other compounds, affecting overall lycopene yield. The *crtEb* gene product, lycopene elongase, catalyzes the elongation of lycopene with DMAPP, yielding the acyclic carotenoid flavuxanthin. The cyclization of flavuxanthin to decaprenoxanthin is facilitated by the carotenoid-ε-cyclase, encoded by *crtY_e_* and *crtY_f_*. The MarR-type regulator CrtR, encoded by the *crtR* gene, suppresses transcription from the promoter (P*crtE*) of the *crt* operon (*crtE*, *cg0722*, *crtB*, *crtI*, *crtY_e_*, *crtY_f_*, *crtEb*). By inhibiting the expression of these repressors of carotenoid biosynthesis, a further improvement in lycopene production can be realized *C. glutamicum* [9,10]. Several critical genes, including *crtE* (*cg0723*), *crtB* (*cg0721*), *crtI* (*cg0720*), *crtY_e_* (*cg0719*), *crtY_f_* (*cg0718*), *sigA* (*cg2092*), *idi* (*cg2531*), and *crtEb* (*cg0717*), which play significant roles in lycopene or precursor production pathways in *C. glutamicum*, have been extensively studied by previous researchers (Table 1). But some potential genes remain unexplored. Recently, a putative membrane protein, Mmpl, encoded by the *cg0722* gene, has been reported. This protein is classified under the RND (resistance, nodulation, and division) superfamily [11]. It has been implicated in the transport of trehalose monocorynomycolate (TMCM), an essential component of the cell wall, across the inner membrane. Additionally, it is also involved in the biosynthesis of glycolipids like sulfolipids, polyacylated trehalose, and other complex lipids. It was speculated that the overexpression of *cg0722* improved the stress resistance in the fermentation process and enhanced cell growth, further improving the lycopene production. Here, we overexpressed *cg0722* and investigated its effect in lycopene production. Apart from the challenges in identifying essential genes for lycopene synthesis, the scarcity of efficient editing tools is another factor contributing to the understudied lycopene production in *C. glutamicum*. Hence, more powerful and high-throughput gene editing tools urgently need to be explored.

With the development of synthetic biology and metabolic engineering, CRISPR/Cas systems have been explored as a leading-edge genome manipulation technology in numerous organisms, thanks to easy-to-design, high specificity, and high functional genome-editing efficiency. The CRISPR/Cas9 and CRISPR/Cas12a systems stand out as the most extensively utilized among various CRISPR systems [16]. Despite endeavors being made to implement the CRISPR/Cas9 system in *C. glutamicum*, research has been hindered since *C. glutamicum* cannot tolerate the high toxicity associated with *Sp*Cas9 (Cas9 from *Streptococcus pyogenes*) expression [17]. To address this issue, nontoxic *Fn*Cpf1 (Cas12a from *Francisella novicida*) was applied in *C. glutamicum*. But it relies too much on the PAM sequence 5′-TTTN-3′, which limits the number of editable sites on the genome. Moreover, the efficiency of the available CRISPR-based genome editing methods for gene editing remain relatively low, especially for the large gene deletion. It is critical to find efficient nucleases with a broader range of PAM site preferences. MAD7 (also known as ErCas12a) is a Cas12a variant derived from *Eubacterium rectale*, which encodes a monomeric 147.9 kDa polypeptide consisting of 1263 amino acid residues. It has only 31% homology with *As*Cpf1 (Cas12a from *Acidaminococcus* sp.) at the amino acid level, and the effector proteins are smaller than *As*Cpf1. MAD7 has a significant advantage in the editing of large DNA fragments [18]. Although MAD7 has been shown to have significant editing activity in *Japonica rice* [19], *Aspergilli nidulans* [20], *Escherichia coli* (*E. coli*) MG1655 [21], *Bacillus subtilis*, and *Chinese hamster ovary* (*CHO*) cells [22], its application in *C. glutamicum* has not been reported yet.

Here, we reported an efficient genome editing tool for *C. glutamicum* based on the MAD7 nuclease. The optimized CRISPR/MAD7 system was further applied to large fragment (up to 25 kb) deletion in the *C. glutamicum* genome for the first time. In addition, the simultaneous knockout of double gene and site mutation editing was realized. Furthermore, *C. glutamicum* was engineered to increase the intracellular accumulation of the lycopene by blocking the by-products (deletion of *crtEb* and *crtR*) and enhancing the carbon flux (overexpression of *cg0722*, *crtB*, and *crtI*) of lycopene pathways. The optimized CRISPR/MAD7 system provides a new insight into the metabolic engineering of *C. glutamicum* to achieve valuable biomanufacturing.

## 2. Materials and Methods

### 2.1. Strains and Growth Conditions

All the bacterial strains and plasmids used in this study are described in Appendix A. The Top10F’ *E. coli* strain was used as a cloning host for plasmid construction and cultivated at 37 °C in LB medium (10 g/L tryptone, 5 g/L yeast extract, and 10 g/L NaCl), with shaking at 200 rpm. *C. glutamicum* ATCC 13032 was used as the host strain for genome modifications and grown in BHIS medium (37 g/L brain heart infusion and 91 g/L sorbitol) at 30 °C. During the shake flask fermentation of lycopene production, 2 g/L glucose was added, and the flask was shaking at 200 rpm. If necessary, antibiotics were added at the following concentrations: 50 μg/mL of kanamycin, 15 μg/mL of chloramphenicol, 15 μg/mL of tetracycline, and 15 μg/mL of ampicillin for *E. coli*, and 20 μg/mL of kanamycin and 10 μg/mL of chloramphenicol for *C. glutamicum* and its derivatives.

The CGXII minimal medium was used for fed-batch fermentation containing 42 g/L 3-(N-morpholino) propanesulfonic acid, 40 g/L glucose, 20 g/L (NH_4_)_2_SO_4_, 1 g/L KH_2_PO_4_, 1 g/L K_2_HPO_4_, 0.25 g/L MgSO_4_·7H_2_O, 10 mg/L FeSO_4_·7H_2_O, 10 mg/L CaCl_2_, 10 mg/L MnSO_4_·H_2_O, 1 mg/L ZnSO_4_·7H_2_O, 0.2 mg/L CuSO_4_, 0.2 mg/L Biotin, 0.03 mg/L protocatechuic acid, and 0.02 mg/L NiCl_2_·6H_2_O). Kanamycin was added at 25 mg/L when necessary. To create the seed culture, several single colonies were first inoculated into 250 mL Erlenmeyer flasks containing 25 mL of a CGXII medium and incubating at 30 °C with shaking at 200 rpm for 16-18 h. In total, 2% of the seed cultures were transferred to another 250 mL Erlenmeyer flask containing 25 mL of a CGXII medium and cultured under the same conditions for approximately 16-18 h. Furthermore, 2% of the seed culture was inoculated into a CGXII medium in a 5 L fed-batch fermentation (Applikon ez2, GETINGE, Gothenburg, Sweden) containing 2 L of CGXII medium and growth at 30 °C. During the fermentation, pH was maintained at approximately 7.0 by adding NH_4_OH (50%, *v*/*v*), stirring speed of 800 rpm, airflow rate of 10 L/min, and the dissolved oxygen (DO) content was maintained at approximately 20–30%.

### 2.2. Plasmid Construction

All the primers used in this study are described in Appendix A. The primers were designed using SnapGene 4.3.6 and synthesized by Sangon Biotech (Wuhan, China), and the optimized MAD7 sequence was synthesized by BGI Genomics (Shenzhen, China). The amplification of gene fragments and identification of positive transformers were performed by polymerase chain reaction (PCR). The One Step Cloning Kit (Vazyme, Nanjing, China) was used to assemble linear DNA fragments into circular plasmids. The DNA gel Extraction Kit (Tiangen, Beijing, China) was used to concentrate the DNA fragments.

For the construction of pEC-MAD7 plasmid, pEC-XK99E was used as a template to amplify a ~6.9 kb fragment using primers P1/P2 and a ~3.8 kb MAD7 fragment using primers P3/P4. To generate pEC-MAD7-crRNA-TTTA (+), pEC-MAD7 was digested with *Xho*I and *Sac*II and then assembled with a ~0.3 kb fragment which was amplified using P8/P9. To generate pEC-donor-crRNA-TTTA (+), the ~1000 bp left homologous arm and ~1000 bp right homologous arm were amplified by P11-P15, a ~0.5 kb crRNA fragment was amplified using P16/P17, and the ligation of homologous arm and crRNA fragment were assembled using fusion PCR and then integrated into pEC-XK99E through *Xho*I and *Nru*I. For the construction of pEC-MAD7-TTTA (+), the pEC-MAD7-crRNA-TTTA (+) and pEC-donor-crRNA-TTTA (+) were digested with *Sph*I and *Xho*I, respectively, and then assembled. To generate pEC-MAD7-TTTA (−), pEC-MAD7-TTTA (+) was used as a template using P18/P19 and then assembled to become circular plasmids. To generate pJYS1-MAD7-TTTA (+), pJYS1Peftu was used as a template, and a ~6 kb plasmid backbone was amplified using P38/P39 and then assembled with a ~6 kb crRNA-donor-MAD7 fragment that was amplified using P40/P41. pXMJ19-MAD7-TTTA (+) and pBluescript-MAD7-TTTA (+) were constructed using the same method. To generate pJYS1-MAD7-TTTA (−), pJYS1-MAD7-TTTA (+) was used as a template using P51/P52 and then assembled. The construction method used for the pEC and PJYS1 series plasmid containing various crRNAs for gene editing was like the above method. To generate pZ9-*crtE*-*crtB*-*crtI* plasmid, pZ9 was used as a template to amplify a ~5.2 kb fragment using primers P154/P155 and a ~3.7 kb genes fragment using primers P150/P153. The construction method employed for the pZ9 series plasmid was similar to the method described above.

### 2.3. Plasmid Curing for Iterative Gene Editing

*C. glutamicum* containing plasmids of the pJYS1 and pEC series were incubated at 37 °C in BHIS medium without antibiotics for 48 h with shaking at 200 rpm. Then, cultures were diluted and spread onto a BHIS plate and incubated at 30 °C for 48 h for colony picking. A single colony was transferred to BHIS-Kn, BHIS-cm, or BHIS-Kn cm screening plate and incubated at 30 °C for 48 h. Strains that have lost resistance to both Kn and Cm antibiotics simultaneously were selected.

### 2.4. Determination of Cell Growth and Glucose Concentration

The biomass of the strains was determined by measuring OD_600_ using a UV-visible spectrophotometer (V-1300, MACYLAB INSTRUMENT, Shanghai, China). The glucose concentration was determined using a biosensor analyzer (Sieman, Shenzhen, China).

### 2.5. Extraction and Quantification of Lycopene from the C. glutamicum

To extract lycopene from the *C. glutamicum* strains, a 1 mL aliquot of the cell cultures was centrifuged at 10,000× *g* for 5 min [23]. The cell pellet was resuspended in 1 mL of hexane-acetone (3:2, *v*/*v*) buffer with vigorous shaking of the shaker for 60 s. The supernatant with pigment was transferred to an opaque glass tube. The process was repeated three times at 4 °C until both the cell pellet and the supernatant were colourless. The solvent was evaporated at 40 °C, and the residues were dissolved in 1 mL of hexane-acetone buffer. The resulting supernatant was transferred to brown vials with screw caps after filtrating the samples with 0.22 µm nylon filters using a 1 mL syringe. Lycopene quantification was performed by high-performance liquid chromatography (HPLC) with the ThermoU3000 system (Thermo Fisher Scientific, Waltham, MA, USA) equipped with ZORBAX Eclipse, XDB-C18 column, and a UV-Vis detector which detected signals at 450 nm. For analysis, 20 µL of filtered supernatant was used, and lycopene was separated using a binary gradient, transitioning the eluent from 100% methanol-MTBE-water (81:15:4, *v*/*v*) to 100% methanol-MTBE-water (7:90:3, *v*/*v*) over 35 min at a constant flow rate of 1.0 mL·min^−1^ at 40 °C.

## 3. Results

### 3.1. Establishment and Optimization of CRISPR System Based on MAD7 Nucleases

To construct a CRISPR/MAD7 system, we developed an all-in-one plasmid containing MAD7 and crRNA expression cassettes, along with homologous arms. The plasmid chosen for expressing MAD7 was pEC-XK99E, carrying a high-copy replicon *ColE1* and an inducible promoter P*trc* to ensure the efficient expression of MAD7. As MAD7 required a thymine-rich PAM sequence (5′-YTTN-3′) and a single crRNA, we selected TTTA as the PAM site. The crRNA controlled by a strong constitutive promoter, Pj23119, contained a 35 bp direct repeat sequence (5′-GTCAAAAGACCTTTTTAATTTCTACTCTTGTAGAT-3′) and a 21 bp spacer sequence. To assess the lethality of MAD7-induced DBS, *crtI* was selected as the target gene, the knockout of which caused cell colour change from yellow to white (Figure 1a and Appendix A). The expression of the MAD7 protein alone showed no significant toxic effect on cells, and no colony was observed after the introduction of crRNA (Figure 1b). The RecE/T system from the Rac prophage of *E. coli* is known for its highly efficient recombineering ability in *C. glutamicum* [24]. In the absence of RecE/T exogenous recombinase, the provision of 1000 bp repair templates generated few observable transformants, indicating the low efficiency of HDR. Upon introducing the RecE/T system, the number of transformants was significantly improved (Appendix A).

Although the number of transformants increased, it was observed that single gene editing efficiency (positive transformants number/total transformants number) remained less than 20%, with TTTN as the PAM site (Appendix A). To improve the knockout efficiency, we tested different promoters for MAD7 expression based on the pEC-XK99E plasmid. Other than P*trc*, three promoters, including a strong constitutive promoter P*tuf* [25], an inducible promoter P*tac* [17], and a modified lac constitutive expression promoter P*lacM* [26], were investigated, and the editing efficiency of the *crtI* gene was evaluated. The results showed that 5.19%, 0%, 2.89%, and 10.97% editing efficiencies were obtained by P*trc*, P*tac*, P*tuf*, and P*lacM*, respectively. Compared with other promoters, P*lacM* performed best on MAD7 expression. Nonetheless, optimizing the promoter did not result in a substantial improvement in the editing efficiency (ranging from 0% to 10% ± 5%, Figure 1c). Studies have demonstrated the effectiveness of manipulating plasmid copy numbers to advance biotechnological applications and to optimize the behavior of engineered genetic systems [27]. The range of the available copy number for cloning vectors is largely restricted to the handful of Origins of Replication [28]. Therefore, pJYS1Peftu (copy numbers, ~5), pXMJ19 (copy numbers, ~138), pBluescript (copy numbers, 300~500), and pEC-XK99E (copy numbers, 500~700) plasmids with different ORIs were assayed (Appendix A). The results showed that the editing efficiencies with pJYS1Peftu, pXMJ19, pBluescript, and pEC-XK99E were 84.66%, 52.22%, 61.90%, and 10.97%, respectively, demonstrating the fact that a higher copy number of plasmids did not necessarily result in better editing efficiency. These data suggested that, compared with promoters, plasmid with desirable ORIs played a greater role in editing efficiency.

Additionally, *crtI*, *upp*, and *crtEb* were selected to identify the optimal PAM sequence of CRISPR/MAD7. The deletion of the *upp* gene leads to a 5-fluorouracil (5-FU)-resistant phenotype (Appendix A), which facilitates convenient screening. Deleting *crtEb* (encoding lycopene elongase) induces a colour change from yellow to light pink. Various crRNAs were inserted into the pJYS1eftu plasmid to construct a series of plasmids, which targeted different PAM sequences. When the PAM sequences were TTTA or TTTC in targeting *crtI*, the results demonstrated that different PAM site locations (leading strand or lagging strand) had no significant effect on editing efficiency (Figure 1d). Single gene editing efficiency approached 100% when choosing TTTC/G as the PAM site. However, no editing efficiency nor transformants were obtained when the PAM site was TTTT, resembling *Fn*Cpf1 [29]. Moreover, when validating the editing activity of CTTN by knocking out *upp* or *crtEb* genes (Appendix A), CTTN showed a high editing efficiency (85% to 95% ± 5%, Figure 1e,f). This result suggested that MAD7 remained high editing activity when the PAM site was CTTN, which certainly broadens the choice of PAM sites in *C. glutamicum*. In summary, we found that the optimized CRISPR/MAD7 system had remarkable editing activity in *C. glutamicum*.

### 3.2. CRISPR/MAD7 System in Large DNA Fragment Deletion

It has been reported that the CRISPR/*Sp*Cas9 or CRISPR/*Fn*Cpf1 system can delete a 20 kb genomic DNA fragment with an editing efficiency of ~26% in *C. glutamicum* [29,30]. To further evaluate the performance of this editing tool in deleting the fragments of variant sizes, *CGP3*, a 220 kb prophage gene, was chosen as the target to delete 5 kb, 20 kb, and 25 kb DNA fragments in the genome of *C. glutamicum.* Plasmid pJYS1, carrying MAD7 under the control of P*lacM*, along with crRNA featuring PAM sequences of 5′-TTTC-3′, was used to guide MAD7 in cleaving DNA and generating a double strand break (DSB). Approximately 1000 bp left and 1000 bp right homologous arms were used to repair the DSB (Figure 2a). The MAD7-mediated editing efficiency for the 5 kb DNA fragment deletion was an impressive 91.11% (Figure 2b and Appendix A). According to the data from this study, the CRISPR/*Fn*Cpf1 system allowed for 30% editing efficiency when deleting 20 kb DNA fragments. Compared with *Fn*Cpf1, the MAD7 performed better, reaching an editing efficiency of 77.14% (Figure 2c). In addition, the knockout of a 25 kb DNA fragment achieved an editing efficiency of 43.75%, marking the largest successfully knocked-out genomic DNA fragment reported in *C. glutamicum*. These findings demonstrated that, in contrast to the CRISPR/*Fn*Cpf1 and CRISPR/*Sp*Cas9 systems, the CRISPR/MAD7 system offers distinct advantages for knocking out large DNA fragments in the genome of *C. glutamicum*.

### 3.3. Editing Site Range by the CRISPR/MAD7 System

The RecT is an exogenous recombinase, encoded by the *E. coli* Rac prophage, known for promoting ssDNA (single-stranded DNA) homologous recombination at a high frequency [31]. With the aim to explore the capabilities of site mutation using the CRISPR/MAD7 system (Figure 3a), *crtE* was selected as the target gene. The introduction of a stop codon by point mutation causes a colour change in colonies from light yellow to white (Appendix A). Employing 5′-TTTC-3′ as the PAM sequences can achieve the goal of introducing a termination codon change to 5′-TTAA-3′ (Figure 3b). Here, 59 nt and 100 nt oligonucleotide strands were, respectively, used to repair DBS generated by MAD7. The results showed that the 100 nt oligonucleotide strand obtained an editing efficiency of 76.07%, surpassing the 61.04% editing efficiency obtained with the 59 nt oligonucleotide strand (Figure 3c). In addition, we attempted to construct 59 bp of dsDNA for point mutation assisted by RecE/T that obtained 79.46% editing efficiency. According to recent studies, *Fn*Cpf1, *Lb*Cpf1, and *As*Cpf1 [29,32,33] showed editing efficiencies of 91.6%, 91.7%, and 50% at nucleotide mutation, respectively. MAD7 nuclease did not exhibit the highest proficiency in nucleotide mutation when compared to *Fn*Cpf1 and *Lb*Cpf1.

### 3.4. Metabolic Engineering of Lycopene Pathway Assisted by CRISPR/MAD7 System

*C. glutamicum* naturally synthesizes lycopene, but with low yields. It was reported that knocking out specific genes in the bypass pathway can enhance lycopene accumulation, and the knockout of both *crtEb* and *crtY* increased the lycopene production 10-fold in *C. glutamicum* [14]. The *crtEb* gene codes a lycopene elongase that converts lycopene to flavuxanthin, thereby reducing lycopene accumulation. Thus, it was hypothesized that the deletion of *crtEb* could contribute to increased lycopene production. Additionally, *crtR* is located in the divergent region of a carotenoid biosynthesis gene cluster in the *C. glutamicum* genome, but its role in lycopene production has not been investigated. CrtR can repress the transcription of *crt* gene to interfere with lycopene accumulation [34,35] (Figure 4a). Accordingly, it was speculated that deleting *crtR* could relieve the inhibitory effect on the *crt* gene. In this study, the knocking out of *crtR* and *crtEb* generated ∆*crtR* and ∆*crtEb* strains, yielding 0.46 mg/L and 0.54 mg/L of lycopene, respectively, as illustrated in Figure 4b. These strains were cultivated in shake flasks using BHIS medium for 120 h. The knockout of the *crtEb* resulted in a 1.5-fold increase in lycopene production compared to the wild-type strain, with yields reaching 0.37 mg/L. The effect of *crtEb* knockout was more pronounced than *crtR* knockout.

In addition to inhibiting the by-products of the lycopene pathway, manipulating the gene copy number to influence metabolic pathway flux is another strategy used for improving lycopene production [9]. Thus, we increased the copy numbers of key genes, including *crtE*, *crtB*, *crtI*, *idi*, *idsA*, and *cg0722*, to enhance lycopene production. Many novel polycistronic expression cassettes were generated by combining the above different genes. However, before the introduction of new plasmids into *C. glutamicum*, it was necessary to remove the plasmids involved in the CRISPR/MAD7 system with resistance to kanamycin and chloramphenicol. Accordingly, the strains were initially inoculated and cultured in BHIS medium without antibiotics at 37 °C for 48 h. The first round of inoculation only removed kanamycin resistance, and the removal of chloramphenicol resistance required a second round (Figure 4c). After the removal of CRISPR/MAD7-related plasmids, polycistronic expression cassettes of *crtE*-*crtB*-*crtI*, *cg0722*-*crtB*-*crtI*, *crtE*-*cg0722*-*crtB*-*crtI*, and *idsA*-*idi*-*crtB*-*crtI* were introduced into *C. glutamicum*. The polycistronic expression cassettes were overexpressed under the control of the P*tac* promoter (a modified constitutive promoter) in ∆*crtEb* strain, generating EBI, CBI, ECBI, and AiBI strains. These strains achieved lycopene titers of 2.497, 8.688, 7.789, and 8.344 mg/L (Figure 4d), respectively. The results showed that the simultaneous overexpression of *cg0722*, *crtB*, and *crtI* genes can maximize lycopene production. Furthermore, the engineered strains in the present study did not require the addition of IPTG (Isopropyl-β-D-thiogalactopyranoside), mitigating concerns related to high costs and toxicity in large-scale industrial production [36].

As single-gene knockout did not meet expectations for lycopene production, a dual knockout of *crtEb* and *crtR* using the CRISPR/MAD7 system with PAM sequences of 5′-TTTC-3′ was conducted (Figure 5a), achieving an editing efficiency of 20.83% (Figure 5b and Appendix A). The *crtR* knockout resulted in the colony changing from light yellow to dark yellow. Interestingly, the simultaneous knockout of *crtR* and *crtEb* caused a distinct shift in the colony colour from light yellow to pink (Appendix A), indicating a noticeable accumulation of lycopene. The engineered strain, referred to as ∆*crtEb*∆*crtR*, produced 22.6 times more lycopene than the WT and 15.5 times more than ∆*crtEb*, reaching lycopene production of 8.35 mg/L (Figure 5c). By introducing the optimal polycistronic expression cassette *cg0722*-*crtB*-*crtI* into the ∆*crtEb*∆*crtR* strain, an engineered strain named CBIEbR was obtained. Remarkably, the CBIEbR strain produced the highest lycopene yields of 23.12 mg/L, which was 62.5 times higher than that in WT (Figure 5d).

### 3.5. High-Density Fermentation for Lycopene Production

CGXII medium was utilized to demonstrate the feasibility of lycopene production by the engineered strain. Firstly, the WT and CBIEbR strains were inoculated in 250 mL shake flasks using CGXII medium for 120 h of fermentation before fed-batch fermentation. The results show that the lycopene accumulation of the WT strain and CBIEbR strain was 0.3 mg/L and 30.79 mg/L, respectively (Appendix A). Secondly, the lycopene production capacity of the optimal recombinant strain CBIEbR was further assessed through fed-batch fermentation in a 5 L fermenter using a higher glucose concentration (40 g/L) in CGXII medium. To minimize the effect of high glucose concentrations on the osmotic pressure of the strain and to ensure efficient lycopene accumulation, a two-stage glucose feed strategy was implemented. In the first stage (0 h–36 h), glucose was added at a rate of 1.67 g/L/h. In the second stage (36 h–120 h), glucose was added at a rate of 3.34 g/L/h. During fed-batch fermentation, the OD_600_ of the strain reached 228 and accumulated the most lycopene of 405.02 mg/L (9.52 mg/g DCW) at 96 h (Figure 6a). To our knowledge, this is the highest yield reported so far to produce lycopene in *C. glutamicum* (Figure 6b).

## 4. Discussion

*C. glutamicum* is used to produce natural products such as lycopene, which has important economic and social value. Despite the many advantages of lycopene production by *C. glutamicum*, due to the scarcity of efficient genetic tools, there is still little research related to the production of lycopene using *C. glutamicum*. In recent years, a plethora of genetic toolkits have emerged, facilitating the engineering of high-yielding strains. In the case of *C. glutamicum*, traditional genome editing tools, such as Counter-marker-assisted allelic exchange [37] and the Cre-loxP system [38], have been applied with great success. Counter-marker-assisted allelic exchange allows for the precise modification of specific genes by introducing a counter-selection marker, such as the levansucrase gene *SacB*, 5-fluorouracil-lethal gene *upp*, or the streptomycin-sensitive gene *rspl*. Additionally, the Cre-loxP system offers site-specific recombination mediated by the Cre recombinase for the precise excision or integration of DNA segments. Nevertheless, these strategies are time-consuming and inefficient [39]. More recently, nearly all applications have been based on Cas9 or Cpf1. Although *Fn*Cpf1 is not the inherent toxicity and can be used as an alternative to Cas9 for genome editing in *C. glutamicum*, the PAM site preference of 5′-TTTN-3′ limits its applicability. In the present work, we developed an application of MAD7 from *Eubacterium rectale* as an efficient genome editing tool with a broader range of PAM site preferences for the key industrial microorganism *C. glutamicum*.

As a preliminary experiment, we investigated whether the CRISPR/MAD7 editing system functions in the *C. glutamicum*. Unexpectedly, the single gene editing efficiency with TTTN as the PAM site remained less than 20%. Through the optimization of the promoter, ORI, and PAM sequences, the editing efficiency of the CRISPR/MAD7 system improved significantly, from 5.19% to 100%. Moreover, the optimization of the ORI indicated that a low copy number of ORI improves the editing efficiency of the system. According to the report, by screening different PAM sequences such as 5′-NYTV-3′ (where N represents A, T, C, and G; Y represents C and T; and V represents A, C, and G) of the CRISPR/FnCpf1 system, it was found that 5′-TTTA/C-3′ was the preferred design [40]. Therefore, no further optimization was performed in this study when the PAM site was 5′-NYTV-3′. In addition to the promoter, ORI, and PAM sequences, it has been reported that systematically attenuating the DNA targeting activity also benefits CRISPR-driven editing, involving alterations to the format or expression strength of guide (g)RNAs, the use of nucleases with reduced cleavage activity, and engineering-attenuated gRNAs [41]. Moreover, factors such as GC content, nucleotide bias, gRNA length, and homology arm length may also affect the editing efficiency of the CRISPR/MAD7 system, warranting further exploration.

While large DNA fragment deletion tools can accelerate genome evolution, the difficulty of deleting DNA fragments typically increases with their size. Utilizing the CRISPR/MAD7 system, a 25 kb DNA fragment was successfully knocked out in *C. glutamicum*. In contrast, CRISPR/*Fn*Cpf1 has not achieved the successful deletion of a fragment of this size in *C. glutamicum*. Although multiplex gene editing has been accomplished using *Fn*Cpf1, it mostly performs point mutations rather than multiple gene knockouts. Here, the CRISPR/MAD7 system combined with two crRNA arrays simultaneously targeted two genes in one step and obtained 20.83% editing efficiency. Despite the relatively moderate efficiency of double knockout, it holds promise as a high-throughput editing tool for genomes after further system optimization.

The resultant strain had the potential in transition from a lab-scale setup to industry-scale setup. Firstly, we avoided the use of IPTG during the construction of the engineered strains, thus reducing industrial production costs and decreasing the toxic effects on the cells. Although IPTG is a commonly used inducer for industrial microbial production [12], the use of IPTG increases the cost of industrial production and is toxic to cells. Secondly, shorter fermentation time allowed for more water and electricity savings in industrial production. Engineered *C. glutamicum* reached the accumulation of the highest lycopene yield in 96 h, whereas yeast lycopene accumulation took 120 h or more to peak [2]. Thirdly, *C. glutamicum* can utilize a wide range of natural carbon sources like fructose, mannose, acetate, and ferulic acid and can be engineered using other carbon sources such as lactate, xylose, and cellulose [6]. Among the carbon sources, cellulose from agricultural waste is the cheapest one. The use of cellulose from agricultural waste instead of glucose to produce natural products could make microbial synthesis more economically viable.

## 5. Conclusions

In this study, we have crafted a novel genetic editing tool tailored for advancing metabolic engineering in *C. glutamicum*. Through modifications of the host cell and the carotenogenic pathway, the resulting strain exhibited outstanding proficiency in the accumulation of lycopene, leading to a remarkable increase in the lycopene yield to 405.02 mg/L (9.52 mg/g DCW). This work expands the current CRISPR/Cas toolbox for *C. glutamicum* and demonstrates the applicability of this system. The developed tools and strategy can provide a reference for the bioproduction of valuable natural compounds in *C. glutamicum* and other industrial organisms.

## Figures and Tables

**Figure 1 microorganisms-12-00803-f001:**
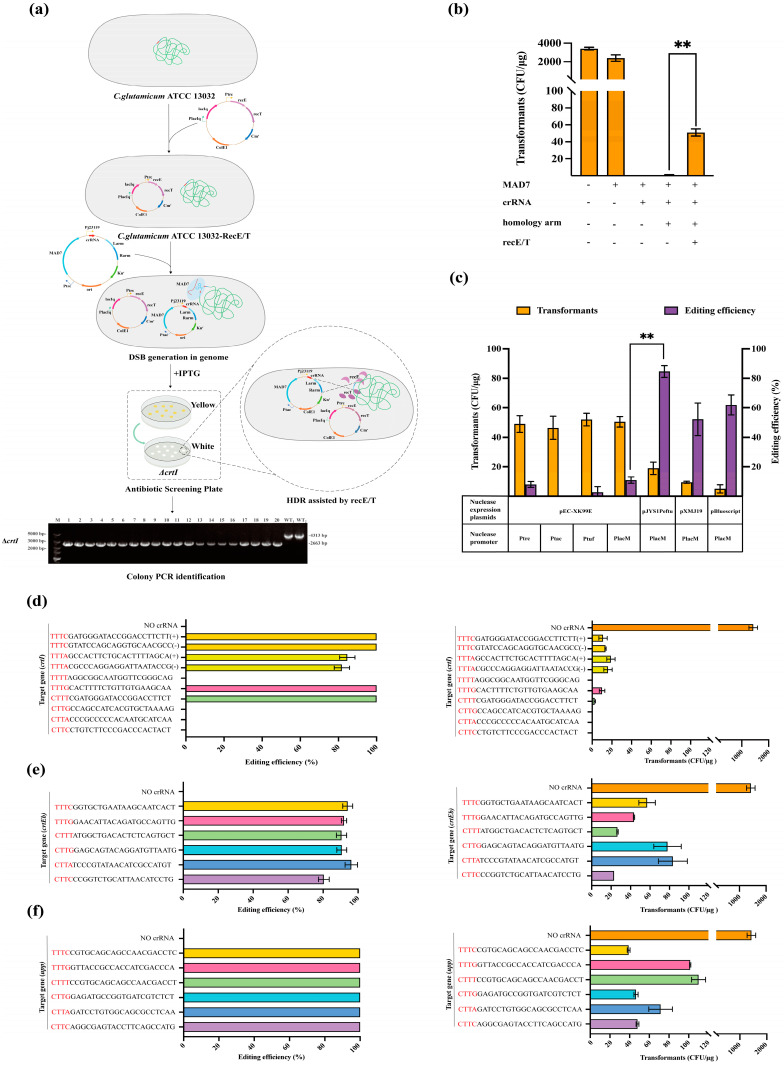
Establishment and optimization of CRISPR/MAD7 system in *C. glutamicum*. (**a**) The single plasmid-based CRISPR/MAD7 system is assisted by RecE/T for genome editing. The pEC series plasmid expressing MAD7, crRNA, and homologous arms with Kn^r^. The RecE/T was carried by a pEC plasmid with Cm^r^ and under the control of an inducible promoter P*trc*. A total of 1 mmol/L of IPTG is used to induce RecE/T and MAD7 expression. (**b**) Transformants of *C. glutamicum* cells expressing the nuclease, with or without the combined expression of RecE/T, homology arm, and crRNA targeting *crtI*. (**c**) Transformants and editing efficiency of MAD7 nuclease under different promoters and expression plasmids targeting *crtI*. (**d**) The effect of the various PAM sequences on the efficiency and transformants of editing the *crtI* (red part represents the PAM sequence). (**e**) The effect of the various PAM sequences on the efficiency and transformants of editing the *crtEb* (red part represents the PAM sequence). (**f**) The effect of the various PAM sequences on the efficiency and transformants of editing the *upp* (red part represents the PAM sequence). Pj23119, a synthetic constitutive expression promoter; P*trc* and P*tac* were inducible promoters; P*lacM*, a modified lac constitutive expression promoter; P*tuf*, a strong constitutive promoter; lacIq, lac repressor of *E. coli*; Cm^r^, chloramphenicol resistant; Kn^r^, kanamycin resistant; IPTG, Isopropyl β-D-Thiogalactoside. CFUs, colony forming units. Data are analyzed using two-tailed *t*-test, ** *p* < 0.01.

**Figure 2 microorganisms-12-00803-f002:**
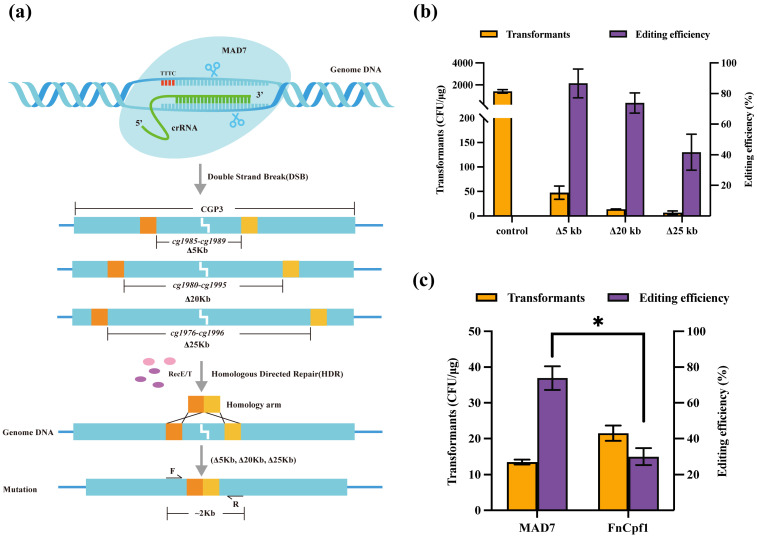
Large DNA fragment deletion by CRISPR/MAD7 system in *C. glutamicum*. (**a**) Schematic representation of 5 kb, 20 kb, and 25 kb genome DNA fragment deletion in *C. glutamicum*. (**b**) Editing efficiency and transformants of 5 kb, 20 kb, and 25 kb genome large DNA fragment deletion by MAD7. (**c**) Comparison of the editing efficiency and transformants of 20 kb genome large DNA fragment deletion by MAD7 and *Fn*Cpf1, respectively. Data are analyzed using two-tailed *t*-test, * *p* < 0.05.

**Figure 3 microorganisms-12-00803-f003:**
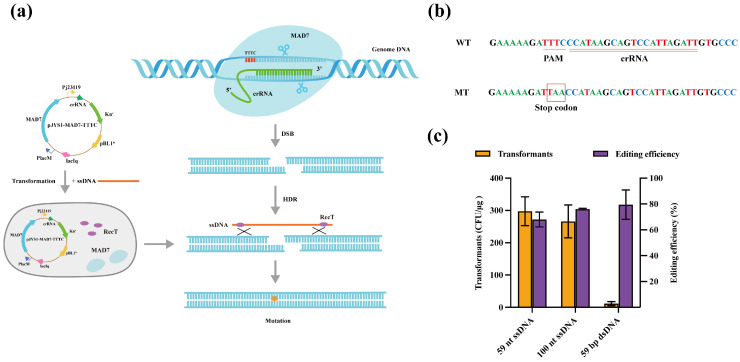
Scheme for site mutation via the CRISPR/MAD7-RecT system in *C. glutamicum*. (**a**) crRNA guiding MAD7 to cleave genomic DNA and produce DSBs that can be repaired by RecT and ssDNA. (**b**) Sequencing results of the *crtE* gene in the WT (*C. glutamicum* ATCC 13032) and MT (*C. glutamicum* ATCC 13032-*crtE* mutation) strains; nucleotides in red box represent the stop codons (TAA) introduced by ssDNA-directed recombineering. (**c**) Number of transformants and editing efficiency of the point mutation by CRISPR/MAD7-RecT system using dsDNA or various ssDNA.

**Figure 4 microorganisms-12-00803-f004:**
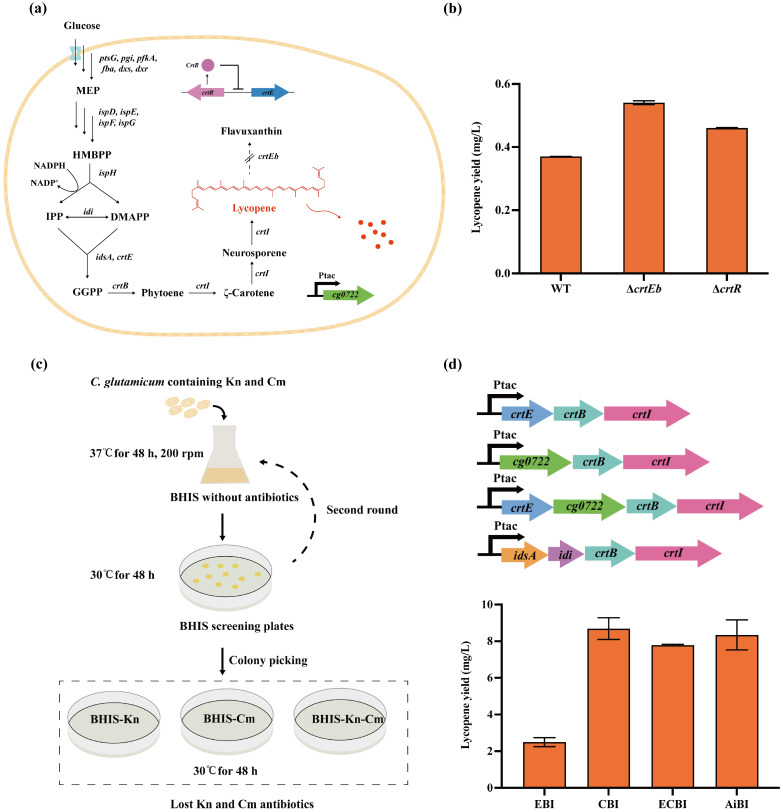
Metabolic engineering of the lycopene pathway through single gene deletion and multiple gene overexpression. (**a**) Scheme of the lycopene biosynthesis pathway in *C. glutamicum*. Lycopene is distributed in lipid structures. (**b**) Lycopene yield of genes knockout using BHIS medium (2 g/L glucose was added). (**c**) A flow-chart for removing genome editing plasmids in *C. glutamicum*. (**d**) Lycopene production in *C. glutamicum*-∆*crtEb* with a combination of various genes using BHIS medium (2 g/L glucose was added).

**Figure 5 microorganisms-12-00803-f005:**
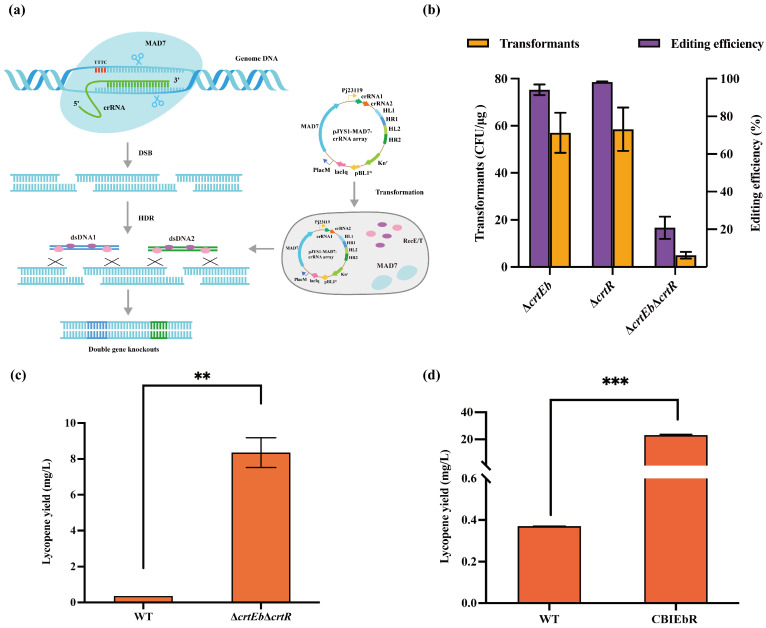
Metabolic engineering of the lycopene pathway through dual gene deletion and multiple gene overexpression. (**a**) crRNA array guiding MAD7 to cleave genomic DNA and produce two DSBs that can be repaired by RecE/T and dsDNA. (**b**) The number of transformants and editing efficiency of the double gene knockout by CRISPR/MAD7 system using dsDNA. (**c**) Lycopene production of ∆*crtEb*∆*crtR* strain using BHIS medium (2 g/L glucose was added). (**d**) Lycopene production of WT strain and CBIEbR strain using BHIS medium (2 g/L glucose was added). Data are analyzed using two-tailed *t*-test, ** *p* < 0.01, *** *p* < 0.001.

**Figure 6 microorganisms-12-00803-f006:**
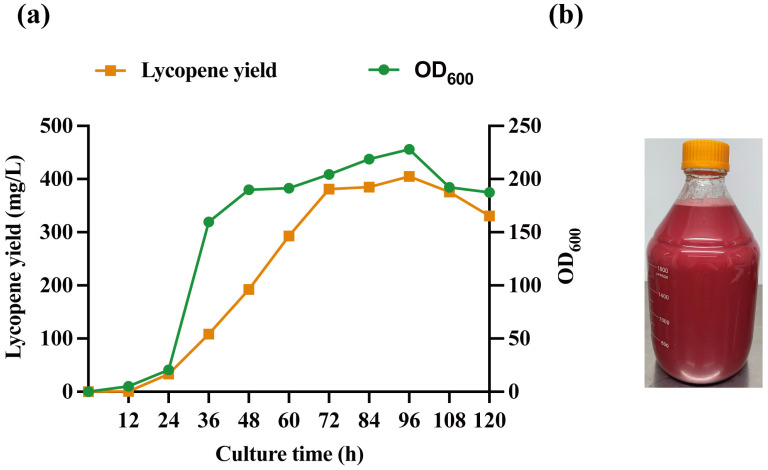
High-Density fermentation for lycopene production. (**a**) Time courses showing changes for CBIEbR strain in lycopene production and cell growth (OD_600_) during fed-batch fermentation. (**b**) The red liquid in the bottles was the lycopene fed-batch fermentation broth.

**Table 1 microorganisms-12-00803-t001:** Strategies for lycopene production in *C. glutamicum*.

Strategy	Titer (mg/g DCW)	Reference
Deletion of *crtEb* and *crtYe*, expression of *crtE*, *crtB*, *crtI* from *C. glutamicum* by changing native promoter into the P*tac* promoter	4.56	[10]
Deletion of *crtEb*, overexpression of *crtE*, *crtB*, *crtI*	2.4	[9]
Overexpression of *sigA*	0.82	[12]
Overexpression of *idi*	0.08	[13]
Deletion of *crtEb* and *crtYe*, expression of *crtE*, *crtB*, *crtI*	0.79	[14]
Deletion of *crtEb*, *crtYe*, and *crtYf*, expression of *crtE*, *crtB*, *crtI*, *dxs* from *C. glutamicum* using a P*tuf* promoter	0.5	[15]

## Data Availability

Data are contained within the article and Appendix A.

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
