# Peer review of "Expanding the CRISPR Toolbox for Engineering Lycopene Biosynthesis in Corynebacterium glutamicum"

_microorganisms, 2024, doi:10.3390/microorganisms12040803_

Round 1

Reviewer 1 Report

Comments and Suggestions for Authors

Dear Authors,

The overall study performed is quite systematic and results obtained are also significant. Please revise the MS  before its consideration for publication.

Title: Establishment of CRISPR/MAD7 gene editing tool and engineering of lycopene biosynthesis in Corynebacterium glutamicum

Comments

1.     The authors have used a powerful and precise genetic tool was used to engineer C. glutamicum for lycopene production. Through the modifications between host cell and carotenogenic pathway, lycopene yield was stepwise improved by 102-fold as compared to the starting strain. This study highlighted the usefulness of CRISPR/MAD7 toolbox, demonstrating its practical applications in the metabolic engineering of industrially robust C. glutamicum. the final engineered strain produced lycopene at 405.02 mg/L in fed-batch fermentation. The manuscript is scientifically sound and well written considering the explanation of the subject.

2.     I suggest to include one table as a comparative explanation of the processes reported to date by other researchers.  

3.     To my opinion, this article can be accepted after minor revision 

Author Response

For research article

Response to Reviewer 1 Comments

1. Summary

Thank you very much for taking the time to review this manuscript. Please find the detailed responses below and the corresponding revisions in the re-submitted files.

2. Questions for General Evaluation

Reviewer’s Evaluation

Response and Revisions

Does the introduction provide sufficient background and include all relevant references?

Yes

Are all the cited references relevant to the research?

Yes

Is the research design appropriate?

Yes

Are the methods adequately described?

Yes

Are the results clearly presented?

Can be improved

The results have been revised.

Are the conclusions supported by the results?

Yes

3. Point-by-point response to Comments and Suggestions for Authors

Comments 1:  I suggest to include one table as a comparative explanation of the processes reported to date by other researchers. 

Response 1: We have added Table 1 to the introduction part and the lycopene production strategies were shown in the Table (Page 3, line 99).

Table 1. Strategies for lycopene production in C. glutamicum

Strategy

Titer (mg/g DCW)

Reference

Deletion of crtEb and crtYe, expression of crtE, crtB, crtIfrom C. glutamicum using a Ptac promoter

4.56

[10]

Deletion of crtEb, overexpression of crtE, crtB, crtI

2.4

[9]

Overexpression of sigA

0.82

[12]

Overexpression of idi

0.08

[13]

Deletion of crtEb and crtYe, expression of crtE, crtB, crtI

0.79

[14]

Deletion of crtEb, crtYe and crtYf, expression of crtE, crtB, crtI, dxs from C. glutamicum using a Ptuf promoter

0.5

[15]

Deletion of crtEb and crtR, expression of cg0722, crtB, crtI from C. glutamicum using a modified constitutive promoter Ptac

9.52

This study

4. Response to Comments on the Quality of English Language

5. Additional clarifications

Reference

  1. Heider SA, Peters-Wendisch P, Wendisch VF. Carotenoid biosynthesis and overproduction in Corynebacterium glutamicum. BMC Microbiol, 2012, 12: 198.
  2. Li C, Swofford CA, Ruckert C, et al. Heterologous production of alpha-carotene in Corynebacterium glutamicum using a multi-copy chromosomal integration method. Bioresour Technol, 2021, 341: 125782.
  3. TANIGUCHI H, HENKE N A, HEIDER S A E, et al. Overexpression of the primary sigma factor gene sigA improved carotenoid production by Corynebacterium glutamicum: Application to production of beta-carotene and the non-native linear C50 carotenoid bisanhydrobacterioruberin [J]. Metab Eng Commun, 2017, 4(1-11).
  4. HEIDER S A, WOLF N, HOFEMEIER A, et al. Optimization of the IPP Precursor Supply for the Production of Lycopene, Decaprenoxanthin and Astaxanthin by Corynebacterium glutamicum [J]. Front Bioeng Biotechnol, 2014, 2(28).
  5. HEIDER S A, PETERS-WENDISCH P, NETZER R, et al. Production and glucosylation of C50 and C40 carotenoids by metabolically engineered Corynebacterium glutamicum [J]. Appl Microbiol Biotechnol, 2014, 98(3): 1223-35.
  6. HENKE N A, HEIDER S A, PETERS-WENDISCH P, et al. Production of the Marine Carotenoid Astaxanthin by Metabolically Engineered Corynebacterium glutamicum [J]. Mar Drugs, 2016, 14(7).

Reviewer 2 Report

Comments and Suggestions for Authors

Thank you for the opportunity to review this manuscript. I find it very interesting and well-structured, presenting the achieved results in a clear and logical manner. The used literature is up to date and appropriate. Nevertheless, here are my suggestions for further improvement of the quality:

1.     First of all, the title should be more attractive to the readers. Try to modernize the existing title or provide a new one.

2.     The sentence in lines 127-128 should be removed since it is part of conclusions not introduction part

3.     Line 341 give exact numbers on lycopene accumulation in the used reference

4.     The discussion part should be revised and more comparisons with available studies should be included. Here are some advices how to do it:

·      According to the obtained results the single gene editing efficiency with TTTN as the PAM site remained less than 20%. In my opinion the authors should discuss about possible ways of optimization and testing of different PAM sequences in order to enhance gene editing outcomes.

·      The paper reveals the possibilities of lycopene production in a lab-scale setup. It is well known that transition from lab setup to industrial scale setup can be challenging. Certain discussion about this process should be provided including technological as well as economical aspects.

5.     In the conclusions part the section about future investigation should be incorporated as well as possible usage of the obtained results since the segment a new perspective for engineering industrial strains is too objective.

Author Response

For review article

Response to Reviewer 2 Comments

1. Summary

Thank you very much for taking the time to review this manuscript. Please find the detailed responses below and the corresponding revisions in the re-submitted files.

2. Questions for General Evaluation

Reviewer’s Evaluation

Response and Revisions

Does the introduction provide sufficient background and include all relevant references?

Yes

Are all the cited references relevant to the research?

Yes

Is the research design appropriate?

Yes

Are the methods adequately described?

Yes

Are the results clearly presented?

Can be improved

The results have been revised.

Are the conclusions supported by the results?

Can be improved

The conclusions have been revised

3. Point-by-point response to Comments and Suggestions for Authors

Comments 1:   First of all, the title should be more attractive to the readers. Try to modernize the existing title or provide a new one.

Response 1: We have revised the existing title Establishment of CRISPR/MAD7 gene editing tool and engineering of lycopene biosynthesis in Corynebacterium glutamicum” to “Expanding the CRISPR toolbox for engineering lycopene biosynthesis in Corynebacterium glutamicum” (Page 1, line 2 and line 3).

Comments 2: The sentence in lines 127-128 should be removed since it is part of conclusions not introduction part

Response 2: We have deleted the sentence “Taken together, our results showed that CRISPR/MAD7 system is an efficient gene editing platform in C. glutamicum”.

Comments 3:  Line 341 give exact numbers on lycopene accumulation in the used reference

Response 3: We have provided exact numbers on lycopene accumulation in the used reference. The revised sentence is “C. glutamicum naturally synthesizes lycopene but with low yields. It was reported that knocking out specific genes in the bypass pathway can enhance lycopene accumulation, and knockout of both crtEband crtY increased lycopene production 10-fold in C. glutamicum [14]”(Page 10, line 340 to line 343).

Comments 4: The discussion part should be revised and more comparisons with available studies should be included. Here are some advices how to do it: According to the obtained results the single gene editing efficiency with TTTN as the PAM site remained less than 20%. In my opinion the authors should discuss about possible ways of optimization and testing of different PAM sequences in order to enhance gene editing outcomes. The paper reveals the possibilities of lycopene production in a lab-scale setup. It is well known that transition from lab setup to industrial scale setup can be challenging. Certain discussion about this process should be provided including technological as well as economical aspects.

Response 4: We have revised two paragraphs in the discussion. Firstly, we discuss the possible optimizations. The revised paragraph is “As a preliminary experiment, we investigated whether CRISPR/MAD7 editing system functions in the C. glutamicum. However, the single gene editing efficiency with TTTN as the PAM site remained less than 20%. Through the optimization of the promoter, ORI, and PAM sequences, the editing efficiency of the CRISPR/MAD7 system improved significantly, from 5.19% to 100%. Moreover, optimization of the ORI indicated that a low copy number of ORI improves the editing efficiency of the system. According to the report, by screening different PAM sequences such as 5’-NYTV-3’ (where N represents A, T, C and G, Y represents C and T, V represents A, C, and G) of CRISPR/FnCpf1 system, it was found that 5’-TTTA/C-3’ was the preferred design [40]. Therefore, no further optimization was performed in this study when the PAM site was 5’-NYTV-3’. In addition to promoter, ORI, and PAM sequences, it has been reported that systematically attenuating the DNA targeting activity also benefits CRISPR-driven editing, involving alterations to the format or expression strength of guide (g)RNAs, the use of nucleases with reduced cleavage activity, and engineering attenuated gRNAs [41]. Moreover, factors such as GC content, nucleotide bias, gRNA length, and homology arm length may also affect the editing efficiency of the CRISPR/MAD7 system, warranting further exploration. ”(Page 14, line 451 to line 466).

Secondly, we have certain discussions about the technological as well as economic aspects of engineering strains. The revised paragraph is “IPTG is a commonly used inducer for industrial microbial production, and Taniguchi et al. used IPTG to induce lycopene production in C. glutamicum, resulting in an 8-fold increase in lycopene accumulation [12]. However, the use of IPTG increases the cost of industrial production and is toxic to cells. We avoided the use of IPTG during the construction of the engineered strains, thus reducing the industrial production costs, and decreasing the toxic effects on the cells. In addition, engineered C. glutamicumreached accumulation of the highest lycopene yield in 96 hours, whereas yeast lycopene accumulation took 120 hours or more to peak [2]. Shorter fermentation times allow for more water and electricity savings in industrial production. C. glutamicum has a wide range of natural carbon sources like fructose, mannose, acetate, and ferulic acid as carbon sources, and can be engineered using other carbon sources such as lactate, xylose, and cellulose [6], and cellulose from agricultural waste is the cheapest source of carbon. The use of cellulose from agricultural waste instead of glucose to produce natural products could make microbial synthesis more economically viable.”(Page 14, line 477 to line 490).

Comments 5: In the conclusions part the section about future investigation should be incorporated as well as possible usage of the obtained results since the segment a new perspective for engineering industrial strains is too objective.

Response 5: We have included the possible utilization in the conclusion and make it less objective. The revised sentence is “This work expands the current CRISPR/Cas toolbox for C. glutamicum and demonstrates the applicability of this system. The developed tools and strategy can provide a reference for the bioproduction of valuable natural compounds in C. glutamicum and other industrially organisms. ”(Page 14, line 496 to line 499).

4. Response to Comments on the Quality of English Language

5. Additional clarifications

Reference

  1. Zhang X, Wang D, Duan Y, et al. Production of lycopene by metabolically engineered Pichia pastoris. Biosci Biotechnol Biochem, 2019, 84: 463-470.

 6.     Becker J, Rohles CM, Wittmann C. Metabolically engineered Corynebacterium glutamicum for bio-based production of chemicals, fuels, materials, and healthcare products. Metab Eng, 2018, 50: 122-141.

12.   TANIGUCHI H, HENKE N A, HEIDER S A E, et al. Overexpression of the primary sigma factor gene sigA improved carotenoid production by Corynebacterium glutamicum: Application to production of beta-carotene and the non-native linear C50 carotenoid bisanhydrobacterioruberin [J]. Metab Eng Commun, 2017, 4(1-11).

14.    HEIDER S A, PETERS-WENDISCH P, NETZER R, et al. Production and glucosylation of C50 and C40 carotenoids by metabolically engineered Corynebacterium glutamicum [J]. Appl Microbiol Biotechnol, 2014, 98(3): 1223-35.

40.   ZHANG J, YANG F, YANG Y, et al. Optimizing a CRISPR-Cpf1-based genome engineering system for Corynebacterium glutamicum [J]. Microb Cell Fact, 2019, 18(1): 60.

41.     Collias D, Vialetto E, Yu J, et al. Systematically attenuating DNA targeting enables CRISPR-driven editing in bacteria. Nat Commun, 2023, 14: 680.

Round 2

Reviewer 2 Report

Comments and Suggestions for Authors

The authors have addressed all my comments for the improvement of paper quality, therefore, I suggest to accept this manuscript in present form.